# Chemical Characterization of Capsule-Brewed Espresso Coffee Aroma from the Most Widespread Italian Brands by HS-SPME/GC-MS

**DOI:** 10.3390/molecules25051166

**Published:** 2020-03-05

**Authors:** Veronica Lolli, Animesh Acharjee, Donato Angelino, Michele Tassotti, Daniele Del Rio, Pedro Mena, Augusta Caligiani

**Affiliations:** 1Department of Food and Drug, University of Parma, 43124 Parma, Italy; michele.tassotti@studenti.unipr.it (M.T.); daniele.delrio@unipr.it (D.D.R.); pedro.mena@unipr.it (P.M.); augusta.caligiani@unipr.it (A.C.); 2Institute of Cancer and Genomic Sciences, Centre for Computational Biology, University of Birmingham, B15 2TT, UK; a.acharjee@bham.ac.uk; 3Institute of Translational Medicine, University of Birmingham, B15 2TT, UK; 4NIHR Surgical Reconstruction and Microbiology Research Centre, University Hospital Birmingham, Birmingham B15 2WB, UK; 5Faculty of Bioscience and Technology for Food, Agriculture and Environment, University of Teramo, 64100 Teramo, Italy; dangelino@unite.it; 6School of Advanced Studies on Food and Nutrition, University of Parma, 43124 Parma, Italy; 7Department of Veterinary Science, University of Parma, 43124 Parma, Italy

**Keywords:** aroma, espresso coffee, capsules, HS-SPME/GC-MS

## Abstract

Coffee capsules market is on the rise as it allows access to a wide selection of coffee, differing in taste and brand. However, few data about the chemical characterization of the capsule-brewed coffee aroma are available. In this work, an untargeted approach using headspace solid-phase microextraction (HS-SPME) coupled to gas chromatography–mass spectrometry (GC-MS) and combined to chemometrics was performed to study and compare aroma profile from 65 capsule-brewed espresso coffees (ECs) commercialized by five of the most representative brands in Italy. Volatile profiles obtained from ECs were subjected to multivariate statistical analysis, which generally did not show a significant variability among coffees belonging to the same brand, except for those modified after the addition of specific flavor additives or aromatic substances (such as caramel, chocolate, etc.). Similarities may be related to the starting coffee brew or the processing method, which is likely the same for each individual brand. Additionally, partial least squares discriminant analysis (PLS-DA) showed that capsules from a specific brand contain the highest concentration of pyrazines, thus characterized by an intense and characteristic aroma, and a stronger note than those from the other brands. This study supports that the chemical analysis in conjunction with chemometric tools is a useful approach for assessing flavor quality, even if the need remains to identify volatile markers of high-quality beverages.

## 1. Introduction

Coffee is the second most popular beverage in modern society after water, and its consumption is continually increasing with a market value close to 21 billion USD [1,2,3]. The popularity of coffee is mostly due to its stimulating effect from caffeine, as well as its pleasant taste and aroma [4]. Coffee can be prepared in many ways but one of the most widespread and known is the Italian “Espresso coffee” (EC). In fact, according to the latest data from the International Coffee organization (ICO) [3], about 1.5 billion cups of coffee a day are consumed worldwide with a growing preference for espresso when compared to other coffee beverages [5].

EC is prepared in a clear and basic way, where a jet of a limited amount of hot water under high pressure passes in a very brief time through a ground and tamped coffee cake [6]. This process produces a concentrated and intensely flavored brew covered by a dense foam creamy layer. Nowadays, several brewing methods are used to make EC [6]; among them, bar and/or home coffee machines and operating with pre-packed doses containing pre-measured ground coffee as pods and capsules, characterized by mixes of coffee cultivars, roasting degrees, and production countries, as well as by the occasional presence of aromas or additives [7]. Being so easy to prepare and enjoy, the popularity of espresso capsules has exponentially grown, together with the number of private label players entering the market and the interest of consumers for the quality of the final product [8]. Some capsules from specific brands were considered to taste better than others, but overall there is a lack of labelled information, so it is difficult to select the optimal choice for the consumer. In fact, no information (or incomplete information) is labelled about the origin and/or cultivar of the coffee blend, so it is very difficult to discriminate the quality of the product based on the starting blend. Furthermore, there are few studies in the literature concerning the analysis of capsule-brewed EC with respect to chemical characterization of the aroma.

As volatile compounds are associated with coffee flavor and consequently impact its acceptance, coffee flavor research has focused on the identification of active aroma compounds, testing multiple sample preparation advanced techniques, as solvent-assisted flavor evaporation (SAFE) and headspace solid-phase microextraction (HS-SPME) combined with untargeted analysis approaches, using gas chromatography-olfactometry [9,10].

In this context, more than 900 volatile components with a large variety of functional groups have been identified in roasted coffee [11,12,13] and classified by chemical families, including furans, pyrazines, ketones, pyrroles, phenols, hydrocarbons, acids and anhydrides, aldehydes, esters, alcohols, sulfur compounds, and others. It is generally recognized that the coffee aroma is not produced by a single compound, but by a multitude of volatiles in some specific proportions. Although the volatile profile of coffee is very complex, bioactive substances (called key odorants) have been identified as important contributors to the taste and correlated to coffee aroma, and consequently to the quality of the final product. Many published papers concern the relationship between the aroma profile of coffee and the chemical composition of the seeds, including other parameters such as the species and cultivars of coffee, climate, and processing methods, such as roasting, storage and brewing methods [14,15,16]. Thus, the chemical composition of the aroma is an important quality marker for EC.

Moreover, HS-SPME/ gas chromatography–mass spectrometry (GC-MS) combined with chemometric multivariate analysis could provide an accurate approach for the analysis of type and concentration of volatile components for the quality assessment [9].

This work explored the possibility to find some peculiar characteristic among products belonging to different commercial brands and to define characteristics that could permit to discriminate based on the chemical aroma profile. To this aim, the chemical aroma profile of 65 different ECs prepared with coffee capsules belonging to the five most common brands in Italy (in accordance with specific manufacturers’ instructions and specific machines) was characterized by HS-SPME/GC-MS, and a relationship between their aroma profile and flavor was assessed, based on their commercial brands and concentration of selected key odorants by the application of chemometric tools (i.e., multivariate statistical methods).

## 2. Results

### 2.1. Volatile Compounds in Espresso Coffees (EC) Brewed by Capsules

More than 70 volatile compounds were detected and grouped by chemical class, including 7 furans, 12 pyrazines, 9 aldehydes, 10 ketones, 5 pyrroles, 9 esters, 2 pyridines, 2 sulfur and 6 phenolic compounds, 3 terpenes, and others, as shown in Table 1. The identified volatiles in the analyzed samples agreed with those reported in other studies for EC [1,11,15,16,17,18,19,20,21].

A semi-quantification of EC volatiles was performed (using the internal standard) and their relative mean and range (min-max) concentrations were expressed as µg/L for samples grouped by brand and reported in Table 2.

These data (Table 2) suggest that the general aroma of ECs from brand D are characterized by a significant higher concentration of volatiles (ANOVA, *p* < 0.05) belonging to different chemical class. 

Distribution of volatile compounds by chemical class for each brand is shown in Figure 1. 

In order to better understand which compounds may influence the EC aroma and the final flavor, the overall percentage variability (%CV) referring to the mean total sum of volatile compounds grouped by chemical class and between samples belonging to the same brand was calculated (Table 3).

Inter-brand variability reported in Table 3 resulted falling in the range between 20–75%. This high variability in volatiles content among capsules belonging to the same brand may suggest a wide selection of differently tasting products. Pyrazines and sulfur compounds (especially for brand A and brand D) resulted in the chemical class showing the highest variation, while the lowest variation was found for pyrroles (in particular, for brand C).

Nevertheless, due to the complexity of EC aroma profile and the large dataset considered, it is difficult to correlate the general volatile fraction with aroma profile of EC and final quality.

Therefore, selected key odorants responsible for coffee flavor were investigated, combining previous literature data and chemometrics. Among these compounds, eight were identified as key odorants for coffee aroma according to [1,18] (i.e., 2-methylpropanal, 2-methylbutanal, 3-methylbutanal, 2,3-pentanedione, 2-ethylpyrazine, 2-ethyl-6-methylpyrazine, 2-ethyl-3,5-dimethylpirazine and guaiacol) and listed in Table 4, together with their odor description and contribution on flavor [1,15,16,17]. These substances have been suggested to have either a positive or a negative contribution on flavor, and therefore a correlation with the aroma profile of EC [18].

### 2.2. Selection of Volatile Markers for EC Samples Classification Among Brands

#### 2.2.1. Principal Component Analysis (PCA)

Principal component analysis (PCA) was used to obtain an easily interpretable qualitative description of the eventual differences existing in the aroma profile of EC and, in particular, for ECs belonging to the same brand. PCA was performed on the matrix of 74 volatiles for 65 ECs brewed by capsules, and the principal components (PCs) were constructed with the correlation method. Three PCs were extracted, accounting for 94.1% of the variation in the volatile fingerprint. Plotting the first two components only (Figure 2A), no specific grouping was evidenced, unless for slightly variations for those samples likely added with specific flavor additives or aromatic substances (such as caramel, chocolate, etc.), which showed PC1 negative values. In addition, variables on PC2 axis contributed to a separation of some samples from brand D (differing in roasting degree) which showed the highest variability. On the other hand, although PC2 and PC3 explained only 7.9% and 1.8% of the total variability respectively, they were plotted in a 2D score plot (Figure 2A) and two groups were slightly evidenced, especially on the PC3 axis. Variables with positive coefficients on PC3 (Appendix A) resulted to be pyrazines and furfural (and its derivatives) (Figure 2B), responsible for imparting characteristic flavor notes on coffee brew [12] and to group samples belonging to brand D from the others. However, samples of brand D were difficult to classify, probably because they presented the highest inter-brand variability (Table 3), so they exhibited well-defined characteristics for each type of capsule.

Globally, results suggested that EC brewed by capsules belonging to the same brand may present similar characteristics, except for those belonging to brand D, characterized by a high variability.

#### 2.2.2. Partial Least Squares–Discriminant Analysis (PLS-DA)

All the metabolites were included in a partial least squares discriminant analysis (PLS-DA) model to additionally screen the chemical composition for capsule-brewed EC aroma according to the brand. Discriminant markers selection was performed merging the metabolites resulting from the PLS-DA loadings plots (Figure 3A) with those obtained using the Variable Influence in Projection (VIP threshold > 1), shown in Figure 3B.

PLS-DA selected 15 compounds as statistically significant volatiles (VIP > 1), including 9 pyrazines (2,6-dimethylpyrazine (MET37), 2-methylpyrazine (MET39), 2,5-dimethylpyrazine (MET40), 2-ethylpyrazine (MET 41), 2,3-dimethylpyrazine (MET42), 2-ethyl-6-methylpyrazine (MET43), 2-ethyl-5-methylpyrazine (MET44), 3-ethyl-2,5-dimethylpyrazine (MET48) and 2,6-diethylpyrazine (MET49)), furfuryl formate (MET18), 1-(furan-2-yl)ethenone (MET27), 1-methyl-1H-pyrrole-2-carbaldehyde derivative (MET55), 1H-pyrrole derivative (MET 57), and two unidentified compounds (MET47 and MET59). In order to simplify data interpretation on the chemical aroma profile, only pyrazines were considered for further evaluation among the volatiles with a high VIP score.

Globally, results highlighted that ECs from brand D contain the highest content of pyrazines in comparison to the other brands (Figure 4), supporting previous results obtained using ANOVA (*p* < 0.05) (see Table 2) and PCA (Figure 2), and as discussed below. 

## 3. Discussion

In this work, the chemical aroma profile of ECs prepared from capsules from the five most representative brands in Italy were used for classification based on their commercial brands and selected key odorants, by the application of multivariate statistical methods.

Overall, results showed that the qualitative aroma chemical profiles were similar for all the 65 ECs brewed from capsules, especially within the same brand, with some exceptions discussed in detail in the following subsections. The discussion and interpretation of the results in this work highlights the main chemical families, which characterize the EC volatile fraction and contribute to the aroma and final quality of coffee.

### 3.1. Aldehydes and Ketones

2-methylpropanale, 2-methylbutanale, and 3-methylbutanale are the Strecker degradation products of branched amino acids (valine, isoleucine, and leucine) and they have been suggested to be key odorants and responsible of malty and fermented flavor [1]. The concentration values of these aldehydes (Table 2) are in good agreement with previous literature data [16] (considering that volatiles are diluted in water) and differ among the type/taste of capsules. However, there was no consistent trend in aldehydes content which could indicate a potential difference in the quality and the perceived aroma from capsule-brewed ECs, especially from different brands.

The nine ketones detected in the analyzed samples have been previously shown to impart buttery and creamy notes to coffee [1,16,18,19] and are present at significant higher concentrations (ANOVA, *p* < 0.05) in the ECs from brand D, and also for 2,3-pentanedione (previously described as key odorant). On the contrary, ECs from brand E contain the lowest level of this compound. However, the detected ketones in this work are present at moderate concentrations in all samples (Table 2) comparing to literature data [19]; thus, it is difficult to explain any aroma differences among them.

### 3.2. Pyrazines and Furans

According to data from the literature, pyrazines and furans are not only the major compounds in terms of concentration, but also the main classes contributing to coffee characteristic aroma, through their impact on flavor, imparting earthy, musty, woody, and papery notes [23].

In this work, 12 pyrazines were identified and among these, 2-ethylpyrazine, 2-ethyl-6-methylpyrazine and 2-ethyl-3,5-dimethylpyrazine have been previously indicated as potent key odorants [23,24,25,26]. The concentration pattern of pyrazines semi-quantified in this work resulted statistically different among different brands (*p* < 0.05) and the selected 9 pyrazines were highlighted in the PLS-DA as statistically important variables (VIP > 1). Along with thiazoles, pyrazines have the lowest odor threshold, therefore they significantly contribute to the coffee aroma. Moreover, structure-related alkylpyrazines from coffee may exhibit similar properties and contribute to the diverse physiological action of coffee [17].

A previous study [12] suggested that high concentration values of pyrazines are related to species and cultivars, mainly roasted powder and brews of Robusta coffee, as 2,3,5-trimethylpyrazine, 2,3-diethyl-5-methylpyrazine, 2-ethyl-5-methylpyrazine, and 3-ethyl-2,5-dimethylpyrazine together with other phenolic compounds (e.g., guaiacol and guaiacol derivatives), alkyl- and furfuryl- pyrroles, pyridine and N-methyl-2-pyrrole-carboxaldheyde. In this study, alkylpyrazines were found at a higher concentration than those previously reported [16] for brewed coffees from green beans. However, these concentrations are lower than those reported for roasted brew coffee [17] both for Robusta and Arabica species, making a classification based on cultivars difficult.

On the other hand, results of PLS-DA showed that capsules from brand D contain the highest number of pyrazines compared to that detected in ECs from the other brands (Figure 4). Among these, 2,6-diethyl-pyrazine belonging to brand D showed the lowest concentration (150.0 µg/mL) in decaffeinated espresso coffee, so comparable to other decaffeinated espressos from the other brands (the lowest value is 70.0 µg/mL from brand A) (Table 2). This result is also in agreement with data reported in the literature for decaffeinate ECs, which generally contain lower concentration of pyrazines, probably due to the decaffeination procedure [17].

As a whole, EC capsules from brand D are characterized by an intense characteristic aroma and a stronger note than those of other brands, especially with respect to brand A and brand B characterized by a finer flavor.

Regarding furans, eight compounds were detected in all the analyzed EC samples, except for vinyl furan, which generally showed the lowest mean concentration range among these compounds (0–80.6 µg/L). In 1995, the International Agency for Research on Cancer (IARC) classified furan as type 2B— “possibly carcinogenic to humans”. In addition, furans, mostly generated upon roasting from the thermal degradation of endogenous components during the Maillard reaction, and their metabolites (i.e., methyl furans), are suggested to contribute to furan toxicity. Despite coffee being a significant dietary source of methyl furans (where 2-methylfuran levels consistently exceed those of furan), in 2016 the IARC completed their reassessment on the potentially carcinogenic effects of coffee, reclassifying it as type 3—“not classifiable as to its carcinogenicity to humans”—due to insufficient evidence [27]. In fact, there are some gaps of knowledge about the toxicity of furan and methyl furans, especially about exposure to them [28].

Amongst numerous factors, furan and its methyl derivatives concentration in the EC cup depend on coffee composition, processing steps, and brewing methods. In a previous study [27], coffee brewed from capsules was suggested to provide the minimum exposure to furan and its derivatives with respect to other brewing methods (except for instant coffee, which exhibited furans levels below the limit of detection), contributing least to consumer’s dietary exposure to furans from coffee [28]. These data are in contrast with those reported by other authors, suggesting that commercial packed coffee capsules showed higher furan and furan derivatives concentrations than those detected in coffee samples from other brewing methods [29].

In this work, no detectable levels of furan were found in EC samples, according to previous data [27]. Regarding furan methyl derivatives, concentrations of 2-methylfuran and 2,5-dimethylfuran ranged from 45.0 to 531.0 µg/L and from 9.0 to 112.0 µg/L (Table 2) for all samples, respectively. It is noticeable that the concentration pattern of methyl furans shows a high variability and, in some cases, results higher than those reported by [27], however, they fall in the range reported by others [28,29]. An interesting result is that samples, mainly ECs from brand A and brand B, showed a significant variability among methyl furan derivatives concentrations. Among these, the samples added to specific flavor additives or aromatic substances (such as caramel and chocolate) resulted to be characterized by the highest levels of methylfurans, especially 2,5-dimethylfuran, which exceeded the reported ranges by [27]. The presence of these flavor additives known to contain significant level of furans [28] could have contributed to increase their concentrations in the coffee brew. On the contrary, capsules-brewed ECs from brand C and brand E show significant lower content of methyl furans than those from the other brands, especially for 2,5-dimethylfuran (21.0 µg/L and 14.3, respectively).

Overall, these results provide further data about methyl furans content in ECs brewed from capsules and might be useful to for the evaluation of furan and its derivatives’ exposure.

### 3.3. Phenolic and Sulfur Compounds

Among phenolic compounds, guaiacol was detected in all the samples with a concentration range of 210.0–3010.0 µg/L and classified as a key odorant with a spicy and phenolic note [12]. However, no statistically significant differences in the number of phenolic compounds, defined as key odorants, were found among brands (ANOVA, *p* ≥ 0.05), indicating that, in this work, concentration patterns of these volatiles had scarce significance for discriminating aroma chemical differences.

Sulfur-containing compounds were the most likely group to undergo changes during the analysis, probably due to evaporation, oxidation, degradation, and/or interactions with the brewed coffee matrix [16].

Fresh roasted coffee generally contains very low amounts of mercaptans, but during storage they increase considerably [12].

In this study, especially dimethylsulfide (not quantifiable) and 2-(methylsulfanylmethyl)furan were detected in most of the analyzed ECs (Table 2), in accordance with data reported in the literature [12,30,31]. The high variability in their concentration (Table 3) is possibly due to the species, the different way of processing (drying/roasting) and storage conditions, including packaging.

Robusta is known to contain more 2-(methylsulfanylmethyl)furan that Arabica [12]. Concentrations of this volatile in ECs from capsules of brand B (mean 306.3 µg/L) and brand D (mean 730.0 µg/L) significantly differed (ANOVA, *p* < 0.05). However, both concentration ranges for this compound fitted with those measured for Arabica and mixtures obtained in previous studies [30], thus not being able to discriminate between species, chemical aroma profile, and quality differences.

## 4. Materials and Methods

### 4.1. Materials

Capsules from sixty-five different types of coffee capsules belonging to five different brands (named A to E), together with their relative coffee machines, were purchased on local markets in Parma (Italy) or through online stores during 2016 (23 capsules from brand A; 15 capsules, brand B; 10 capsules, brand C; 10 capsules, brand D; and 7 capsules, brand E). Details on the type of coffee are fully described in reference [7].

Toluene, 2-methylbutanal, 3-methylbutanal, 2,3,5-trimethylpyrazine, and 1H-pyrrole-2-carbaldehyde were purchased from Sigma-Aldrich (St. Louis, MO, USA).

### 4.2. Preparation of Espresso Coffee

Samples were prepared as previously described [7]. Briefly, ECs were brewed with bi-distilled water by using the coffee capsule with its relative brand machine, according to the manufacturer’s instructions in terms of coffee volume and extraction time. Aliquots of the brewed coffees (3 mL) were placed in a vial for HS-SPME/GC-MS tightly capped with a silicone/Teflon septum cap and stored at −80 °C until analysis, as described in the following.

### 4.3. Determination of Volatile Profile of ECs

#### 4.3.1. Headspace Solid-Phase Microextraction (HS-SPME)/GCMS

HeadSpace-Solid Phase Microextraction-Gas Chromatography (HS-SPME/GC-MS) was used to determine the volatile profile of EC from capsules, according to [1] slightly adapted. 

Then, 3 mL of coffee sample were added to 10 µL of the standard solution of toluene in H_2_O at 0.3 mg/mL. Samples were analyzed using a HT2800T autosampler (HTA srl, Brescia, Italy) connected to a GC-MS system (Thermo Trace 1300 gas chromatograph (Thermo Fisher Scientific, Waltham, Massachusetts, USA) coupled to a Thermo Scientific ISQ GC-Single Quadrupole Mass Spectrometer (Thermo Fisher Scientific). Sample was equilibrated at 60 °C for 5 min; then, a PDMS fiber (SUPELCO, Bellefonte, PA, USA) was performed for the extraction (25 min) at the same temperature with shaking. 

Volatile compounds were desorbed for 3 min and injected into the GC-MS system by splitless injection (250 °C; 2 min valve delay). Separation was performed on a SUPELCO-Wax column (SUPELCO; 30 m × 250 μm i.d., film thickness 0.25 μm). Helium was used as the carrier gas at a constant flow rate of 1 mL/min. Oven temperature program was isothermal for 3 min at 40 °C, raised to 200 °C at a rate of 3 °C min^−1^ and maintained at 200 °C for 5 min. The mass spectrometer operated in the electron impact (EI) ionization mode (70 eV) with a scan range of 40–400 *m/z*. The ion source temperature was set at 230 °C. All analytical samples were randomized for GC-MS analysis.

Fiber blanks were run daily to ensure the absence of contaminants or carry-over. Before use, all fibers were conditioned as recommended by the manufacturer, and tested to evaluate the consistency of their performance versus a reference EC coffee sample.

#### 4.3.2. Identification of Volatiles in ECs

Compounds were identified based on their retention indices (RI) on the SUPELCO Wax column, their mass spectra, and the injection of pure standard when available. A series of *n*-alkanes (C8–C30) was analyzed to establish RI, as Kovats Indices, which were compared with data from the literature. The mass spectra were compared with those from NIST 11 library. Volatile and identified coffee compounds were grouped by chemical families (furans, pyrazines, pyrroles, pyridines, esters, aldehydes and ketones, among others, being terpenes, sulfur, and phenolic compounds).

The relative concentration of each identified volatile substance was defined by manually integrating its peak area and calculating with respect to the internal standard, and expressed as µg/L.

### 4.4. Statistical Analysis

Data were collected with Xcalibur 2.2 SP1 w Foundation 2.0 SP1. Data set was normalized by auto scaling (x-mean(x)/S.D.(x)) for further statistical analysis. Unsupervised PCA and supervised PLS-DA models were applied. PCA was applied as a preliminary analysis to better understand the relationships among individual ECs. PLS-DA model was fitted using brand as an outcome or response variable and volatile compounds as predictor set.

To quantify the rank of the volatile compounds, Variable Importance in Projection (VIP) scores were used [32]. A variable with a VIP score greater than 1 can be considered significant/important in the model. Analysis was done using R (v3.6.1) and Metaboanalyst’s output [33]. Model statistics were quantified based on the fraction of the sum of squares for the selected component (R^2^), which equates to the percentage of the model variance explained, and the predictive ability (Q^2^). Cross-validation was performed to predict and estimate the model performance (whether models were over fitted). For PLS-DA models, random permutation was used, whereby the class membership of individual samples was permuted randomly.

ANOVA and post hoc Tukey test (*p* < 0.05) were performed using SPSS statistical software (Version 21.0, SPSS Inc., Chicago, IL, USA) to evaluate significant differences in selected volatile concentrations among groups (i.e., ECs belonging to different brands).

## 5. Conclusions

HS-SPME/GC-MS analysis of EC aroma is a simple, rapid, sensitive, and high reproducible method, because the direct sampling does not require further operations (such as extraction) and provides the detection of a wide number of volatiles, giving the chemical fingerprint of the investigated samples. Results from multivariate statistical analysis showed that, based on their chemical aroma profile, all the analyzed ECs brewed from capsules could not be differentiated by specific sensory notes and according to the profiling of the products; except for capsule-brewed ECs from a specific brand (brand D), that are generally characterized by a stronger note if compared to others, characterized by a finer flavor.

Moreover, results may indicate that there is a characteristic starting coffee powder for each different brand, which can be slightly modified by adding coffee powders of specific species, origin, and roasting.

However, the addition of additives or aromatic substances (e.g., caramel, chocolate, vanilla, etc.) may obviously be responsible for the sensory attributes of some capsules and might be used to increase the variety of EC types/tastes. The relationship between the aromatic profile and the high quality of drinks, together with volatile markers of good quality, still requires further studies, especially on sensory attributes.

## Figures and Tables

**Figure 1 molecules-25-01166-f001:**
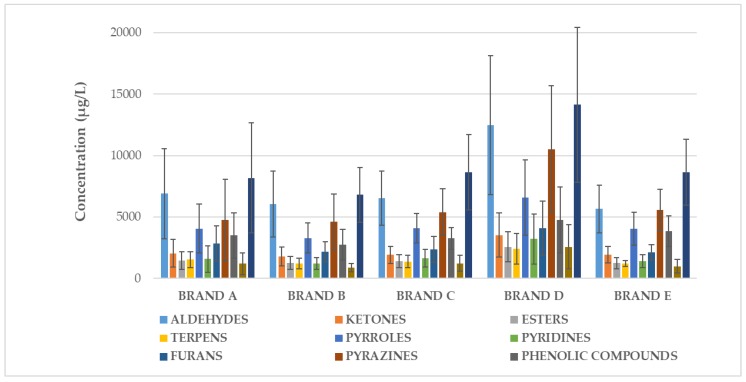
Mean total sum of volatile compounds (µg/L) determined on EC samples from each brand divided by chemical classes. Bars refers to inter-brand variability.

**Figure 2 molecules-25-01166-f002:**
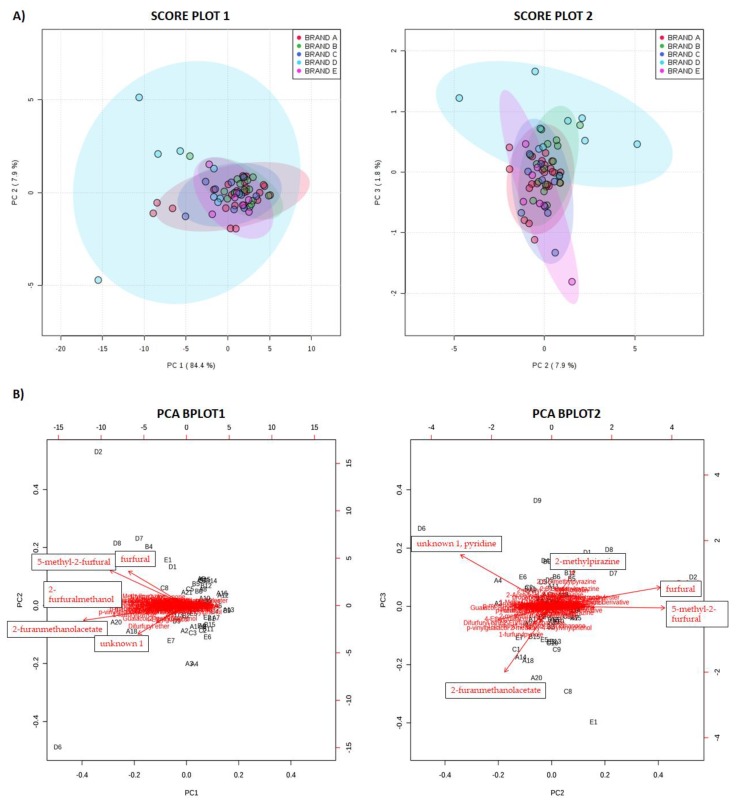
Principal component analysis (PCA) results for volatile profiles in EC brewed by capsules from 5 commercial Italian brands; (**A**) SCORE PLOT1: 2D score plot for PC1 and PC2; SCORE PLOT2: 2D score plot for PC2 and PC3; the ellipse represents the confidence around each of the brands. (**B**) PCA BPLOTS including the position of each sample for PC1 and PC2 (PCA BPLOT1) and for PC2 and PC3 (PCA BPLOT2) and how variables map onto these.

**Figure 3 molecules-25-01166-f003:**
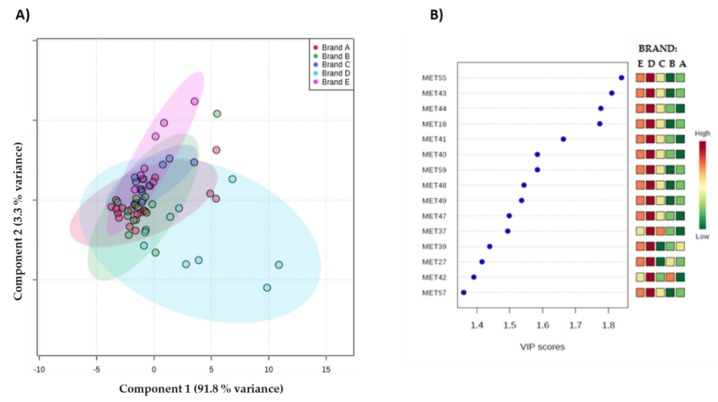
Partial least squares discriminant analysis (PLS-DA) results for volatile profiles in EC brewed by capsules from 5 commercial Italian brands.; (**A**) 2D SCORE PLOT for Component 1 and Component 2. The ellipse represents the confidence around each of the brands; (**B**) VIP score of metabolites resulting from the PLS-DA loading plots with threshold > 1.

**Figure 4 molecules-25-01166-f004:**
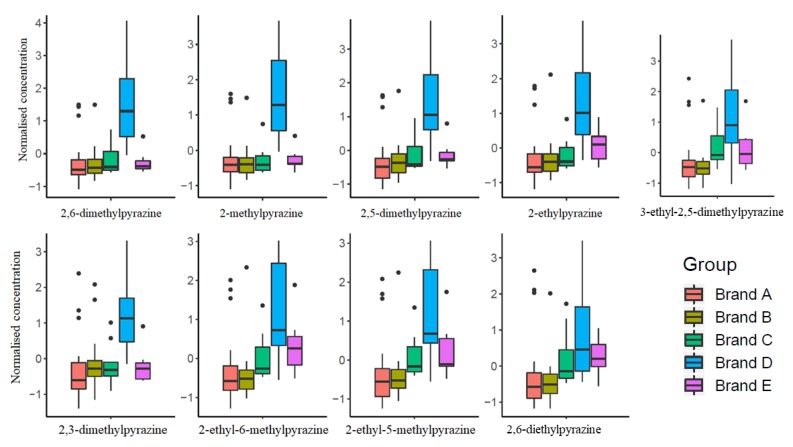
Selected pyrazines by PLS-DA detected in ECs.

**Table 1 molecules-25-01166-t001:** Volatile compounds detected in espresso coffees (EC) samples by headspace solid-phase microextraction (HS-SPME)/gas chromatography–mass spectrometry (GC-MS).

Compound	KI ^1^	ID ^2^	Compound	KI ^1^	ID ^2^
ALDEHYDES			ESTERS		
2-methylpropanal	827	B	methyl acetate	838	B
2-methylbutanal	919	B	ethyl acetate	898	B
3-methylbutanal	921	A	propanoic acid, 3-methyl-, methyl ester	1008	C
furfural	1472	B	methyl isovalerate	1020	C
benzaldehyde	1527	A	butyl butanoate	1229	C
5-methyl-2-furfural	1580	B	prenyl acetate	1256	C
2-methylbenzaldehyde	1596	C	furfuryl formate *	1504	B
2-methyl-3-(2-furyl)acrolein	1682	C	2-oxobutyl acetate	1533	C
1H-pyrrole-2-carboxaldehyde	2034	A	2-furfuryl-propanoate	1602	B
KETONES			PYRAZINES		
2-butanone	906	B	pyrazine	1210	B
2-pentanone	979	B	2-methyllpyrazine *	1266	B
2,3-butandione	982	B	2,5-dimethylpyrazine *	1320	B
2,3-pentanedione	1063	B	2,6-dimethylpyrazine *	1325	B
3-penten-2-one	1127	B	2-ethylpirazine *	1334	B
2,3-hexanedione	1134	B	2,3-dimethylpyrazine *	1345	B
1-(furan-2-yl)ethanone *	1512	B	2-ethyl-6-methylpyrazine *	1385	B
1-(furan-2-yl)butan-2-one	1604	B	2-ethyl-5-methylpyrazine *	1390	B
1-(1-methylpyrrol-2-yl)ethanone	1659	B	2,3,5-trimethylpyrazine	1404	A
			2-ethyl-3-methylpyrazine	1419	C
furfural acetone derivative	1910	C	2,6-diethylpyrazine *	1433	B
			3-ethyl-2,5-dimethylpyrazine *	1441	B
TERPENES			SULFUR COMPOUNDS		
β-linalool	1550	B	dimetilsulfide	1070	B
*p*-menthene monoterpenoid derivative	1650	C	2-(methylsulfanylmethyl)furan (furfuryl methyl sulfide)	1490	B
*p*-menthene monoterpenoid derivative 2	1694	C			
PYRROLES			PHENOLIC COMPOUNDS		
1-methyl-1H-pyrrole	1141	B	2-methoxyphenol (guaiacol)	1867	B
1H-pyrrole derivative *	1801	C	2-methoxy-4-methylphenol	1871	B
1-furfurylpyrrole	1837	B	4-ethylguaiacol	2034	B
2-acetyl-1H-pyrrole	1979	B	2-methylphenol (*o*-cresol)	2092	B
1-methyl-1H-pyrrole-2-carbaldehyde derivative *	2107	C	3-methylphenol (*m*-cresol)	2171	B
			*p*-vinylguaiacol (2-methoxy-4-vinylvinylphenol)	2193	B
FURANS			OTHERS		
2-methylfuran	879	B	unknown 1	1459	-
2,5-dimethylfuran	956	B	benzyl alcohol	1880	C
Vinylfuran	1074	B	2-furanmethanol	1667	B
2-(methoxymethyl)furan	1243	B	2-furanmethanolacetate (furfuryl alcohol acetate)	1542	B
2-butylfuran derivative	1524	C	unknown 2 *	2030	-
furan,2,2′-methylenbis	1614	B	unknown 3 *	2038	-
			furfurylammine derivative	2236	C
difurfuryl ether (2,2′-[oxybis(methylene)]) bisfuran	1992	C	unknown 4	2253	-
			indole	2386	B
			3-methylindole	2450	B
PYRIDINES					
Pyridine	1181	B			
4(H)-pyridine, *n*-acetyl	1724	B			

^1^ Kovats Index (KI) experimentally determined for the SUPELCO^®^-Wax capillary column using homologous series of C8-C30 alkanes; ^2^ Identification (ID). Reliability of identification proposal is indicated by: (A) mass spectrum, retention time, and KI according to standards; (B) mass spectrum and retention index comparing literature data [1,12,15,16,17,18,19,20,21,22]; (C) mass spectrum compared with NIST 11 mass spectral database (pubchem.ncbi). * Key odorants selected by partial least squares discriminant analysis (PLS-DA) (VIP threshold > 1).

**Table 2 molecules-25-01166-t002:** Volatile compounds detected in EC samples grouped by chemical class. Minimum (Min), Maximum (Max), and Mean values (µg/L) are reported for samples belonging to five commercial Italian brands. Different letters (when reported) indicate statistically significant difference (ANOVA, *p* < 0.05); nq = not quantifiable.

	BRAND A (N = 23)			BRAND B (N = 15)			BRAND C (N = 10)			BRAND D (N = 10)			BRAND E (N = 7)		
Compound	Min	Max	Mean	Min	Max	Mean	Min	Max	Mean	Min	Max	Mean	Min	Max	Mean
ALDEHYDES															
2-methylpropanal	50.0	770.0	202.6 ^a,b^	50.0	510.0	158.7 ^a^	100.0	260.0	143 ^a^	130.0	570.0	323.3 ^b^	90.0	240.0	148.6 ^a^
2-methylbutanal	120.0	920.0	400.4 ^a,b^	80.0	1120.0	329.4 ^a,b^	140.0	430.0	225.0 ^a^	180.0	800.0	494.4 ^b^	130.0	370.0	217.1 ^a^
3-methylbutanal	130.0	1690.0	494.3	100.0	980.0	357.5	110.0	510.0	228.0	170.0	760.0	486.7	100.0	350.0	180.0
furfural	580.0	4020.0	1774.3 ^a^	690.0	3890.0	1696.2 ^a^	850.0	2930.0	1772.0 ^a^	1640.0	7500.0	3676.7 ^b^	800.0	2580.0	1385.7 ^a^
benzaldehyde	130.0	920.0	386.1	190.0	880.0	390.0	320.0	730.0	460.0	320.0	1090.0	587.8	330.0	730.0	472.9
5-methyl-2-furfural	910.0	5460.0	2456.5 ^a^	780.0	5440.0	2291.9 ^a^	1480.0	4260.0	2754.0 ^a^	2270.0	8830.0	4786.7 ^b^	1300.0	3980.0	2345.7 ^a^
2-methybenzaldehyde	80.0	750.0	300.4 ^a,b^	70.0	400.0	226.9 ^a^	130.0	560.0	284.0 ^a,b^	250.0	930.0	432.2 ^b^	160.0	320.0	225.7 ^a^
2-methyl-3-(2-furyl)acrolein	250.0	1120.0	601.7 ^a^	160.0	740.0	432.5 ^a^	210.0	740.0	445.0 ^a^	560.0	2040.0	1121.1 ^b^	270.0	530.0	414.3 ^a^
1H-pyrrole-2-carboxaldehyde	60.0	1610.0	297.4 ^a,b^	40.0	350.0	188.1 ^a^	150.0	360.0	229.0 ^a^	210.0	970.0	564.4 ^b^	180.0	430.0	265.7 ^a^
KETONES															
2-butanone	30.0	260.0	100.4 ^a^	40.0	250.0	85.6 ^a^	50.0	140.0	74.0 ^a^	80.0	320.0	173.3 ^b^	50.0	150.0	82.9 ^a^
2-pentanone	10.0	100.0	43.0 ^a,b^	10.0	110.0	34.4 ^a,b^	10.0	70.0	27.0 ^a^	20.0	140.0	61.1 ^b^	30.0	70.0	45.7 ^a,b^
2,3-butandione	10.0	170.0	62.6 ^a,b^	20.0	140.0	55.0 ^a^	20.0	70.0	36.0 ^a^	20.0	180.0	97.8 ^b^	20.0	60.0	34.3 ^a^
2,3-pentanedione	40.0	450.0	135.2 ^a,b^	60.0	280.0	118.2 ^a,b^	50.0	130.0	89.0 ^a^	50.0	400.0	197.8 ^b^	40.0	140.0	75.7 ^a^
3-penten-2-one	0.0	60.0	20.0 ^a^	10.0	30.0	15.0 ^a^	10.0	40.0	18.0 ^a^	30.0	90.0	47.8 ^b^	10.0	20.0	12.9 ^a^
2,3-hexanedione	20.0	150.0	61.7	20.0	580.0	81.9	0.0	70.0	38.0	40.0	130.0	83.3	0.0	50.0	18.6
1-(furan-2-yl)ethanone	170.0	1410.0	597.9 ^a^	150.0	1080.0	566.9 ^a^	220.0	1090.0	574.0 ^a^	590.0	2720.0	1402.2 ^b^	360.0	1190.0	641.4 ^a^
1-(furan-2-yl)butan-2-one	60.0	390.0	163.9	40.0	340.0	151.3	100.0	340.0	173.0	110.0	460.0	241.1	60.0	220.0	148.6
1-(1-methylpyrrol-2-yl)ethanone	260.0	1550.0	680.0	220.0	1130.0	537.5	390.0	1270.0	742.0	0.0	2200.0	933.3	480.0	980.0	695.7
furfural acetone derivative	70.0	420.0	182.6 ^a,b^	30.0	290.0	137.5 ^a^	100.0	230.0	151.0 ^a^	130.0	680.0	294.4 ^b^	120.0	230.0	175.7 ^a^
ESTERS															
methyl acetate	30.0	370.0	107.0 ^a,b^	30.0	290.0	87.5 ^a^	50.0	160.0	77.0 ^a^	100.0	500.0	196.7 ^b^	50.0	210.0	102.9 ^a^
ethyl acetate	50.0	370.0	168.3	50.0	220.0	116.3	40.0	500.0	140.0	50.0	330.0	170.0	40.0	170.0	74.3
propanoic acid, 3-methyl-, methyl ester	60.0	620.0	244.8 ^a^	100.0	630.0	255.0 ^a^	120.0	340.0	234.0 ^a^	240.0	1020.0	564.4 ^b^	100.0	340.0	180.0 ^a^
methyl isovalerate	0.0	30.0	13.5	10.0	30.0	16.3	0.0	20.0	13.0	10.0	40.0	17.8	0.0	20.0	10.0
prenyl acetate	10.0	120.0	51.7 ^a^	20.0	130.0	50.0 ^a^	0.0	100.0	52.0 ^a^	0.0	310.0	138.9 ^b^	20.0	110.0	47.1 ^a^
furfuryl formate	0.0	420.0	128.7 ^a^	0.0	290.0	105.6 ^a^	30.0	300.0	138.0 ^a^	180.0	760.0	468.9 ^b^	50.0	280.0	152.9 ^a^
2-furfuryl-propanoate	320.0	1750.0	752.6	220.0	1400.0	639.4	330.0	1410.0	770.0	580.0	2350.0	1030.0	460.0	1110.0	702.9
TERPENS															
β-linalool	50.0	780.0	287.4 ^a^	80.0	490.0	261.3 ^a^	70.0	580.0	242.0 ^a^	130.0	1430.0	536.7 ^b^	70.0	340.0	184.3 ^a^
p-menthene monoterpenoid derivative	200.0	1290.0	610.0 ^a,b^	140.0	890.0	471.9 ^a^	280.0	970.0	513.0 ^a^	440.0	1700.0	868.9 ^b^	300.0	640.0	452.9 ^a^
p-menthene monoterpenoid derivative 2	180.0	1430.0	642.2 ^a^	130.0	750.0	487.5 ^a^	390.0	900.0	631.0 ^a^	420.0	2250.0	1023.3 ^b^	520.0	620.0	575.7 ^a^
PYRROLES															
1-methyl-1H-pyrrole	40.0	560.0	190.4 ^a,b^	40.0	330.0	141.9 ^a,b^	30.0	240.0	156.0 ^a,b^	120.0	550.0	234.4 ^b^	30.0	200.0	94.3 ^a^
1H-pyrrole derivative	200.0	1320.0	593.0 ^a^	170.0	1070.0	468.8 ^a^	370.0	880.0	596.0 ^a^	470.0	2040.0	1017.8 ^b^	490.0	920.0	675.7 ^a,b^
1-furfurylpyrrole	920.0	4100.0	1847.0 ^a,b^	630.0	2670.0	1506.9 ^a^	1220.0	3040.0	1893.0 ^a,b^	1530.0	5050.0	2525.6 ^b^	900.0	2970.0	1531.4 ^a^
2-acetyl-1H-pyrrole	220.0	1680.0	745.2 ^a^	170.0	1310.0	578.8 ^a^	410.0	1140.0	682.0 ^a^	600.0	3040.0	1385.6 ^b^	580.0	1280.0	858.6 ^a^
1-methyl-1H-pyrrole-2-carbaldehyde derivative	230.0	1580.0	687.4 ^a^	290.0	1410.0	600.6 ^a^	420.0	1300.0	770.0 ^a^	640.0	2340.0	1431.1 ^b^	550.0	1400.0	901.4 ^a^
PYRIDINES															
Pyridine	220.0	3490.0	1200.0 ^a^	360.0	1690.0	901.9 ^a^	580.0	2840.0	1244.0 ^a^	1230.0	7150.0	2568.9 ^b^	320.0	1990.0	994.3 ^a^
4(H)-pyridine, *n*-acetyl	140.0	880.0	386.5 ^a^	130.0	720.0	328.8 ^a^	260.0	540.0	412.0 ^a^	270.0	1160.0	642.2 ^b^	330.0	510.0	420.0 ^a^
FURANS															
2-methylfuran	70.0	530.0	187.8	50.0	480.0	161.9	50.0	280.0	104.0	90.0	430.0	211.1	50.0	180.0	100.0
2,5-dimethylfuran	20.0	90.0	34.8 ^a,b^	20.0	70.0	35.0 ^a,b^	10.0	50.0	21.0 ^a^	30.0	110.0	53.3 ^b^	10.0	30.0	14.3 ^a^
vinylfuran	10.0	80.0	34.3 ^b,c^	10.0	40.0	26.9 ^a,b,c^	0.0	40.0	17.0 ^a,b^	0.0	70.0	37.8 ^c^	0.0	30.0	10.0 ^a^
2-(methoxymethyl)furan	40.0	340.0	123.0 ^a^	50.0	260.0	107.5 ^a^	30.0	210.0	98.0 ^a^	70.0	480.0	246.7 ^b^	60.0	170.0	101.4 ^a^
2-butylfuran derivative	110.0	900.0	347.8 ^a^	120.0	420.0	273.7 ^a^	120.0	560.0	329.0 ^a^	360.0	1350.0	747.8 ^b^	120.0	430.0	215.7 ^a^
furan,2,2′-methylenbis	420.0	1850.0	897.4 ^a,b^	310.0	1260.0	735.0 ^a^	450.0	1430.0	826.0 ^a,b^	770.0	2780.0	1283.3 ^b^	460.0	1200.0	624.3 ^a^
difurfuryl ether (2,2′-[oxybis(methylene)] bisfuran	380.0	2750.0	1213.0	200.0	1730.0	833.1	520.0	2040.0	986.0	860.0	4290.0	1515.6	760.0	1470.0	1055.7
PYRAZINES															
pyrazine	30.0	350.0	136.5 ^a^	60.0	240.0	124.4 ^a^	50.0	210.0	109.0 ^a^	110.0	390.0	263.3 ^b^	40.0	130.0	80.0 ^a^
2-methyllpyrazine	210.0	2110.0	808.3 ^a^	400.0	2030.0	796.3 ^a^	550.0	1510.0	782.0 ^a^	960.0	3580.0	2131.1 ^b^	540.0	1270.0	810.0 ^a^
2,5-dimethylpyrazine	140.0	1290.0	488.7 ^a^	220.0	1350.0	513.7 ^a^	400.0	1010.0	555.0 ^a^	490.0	2220.0	1225.6 ^b^	400.0	950.0	574.3 ^a^
2,6-dimethylpyrazine	130.0	1220.0	468.3 ^a^	240.0	1210.0	480.0 ^a^	350.0	890.0	513.0 ^a^	570.0	2300.0	1255.6 ^b^	360.0	810.0	480.0 ^a^
2-ethylpirazine	90.0	960.0	348.3 ^a^	160.0	1050.0	359.4 ^a^	260.0	680.0	376.0 ^a^	330.0	1510.0	841.1 ^b^	270.0	700.0	454.3 ^a^
2,3-dimethylpyrazine	30.0	350.0	116.1 ^a^	50.0	330.0	142.5 ^a^	70.0	230.0	133.0 ^a^	130.0	430.0	260.0 ^b^	90.0	220.0	128.6 ^a^
2-ethyl-6-methylpyrazine	150.0	1310.0	492.2 ^a^	240.0	1420.0	485.0 ^a^	440.0	1080.0	607.0 ^a,b^	410.0	1660.0	981.1 ^b^	420.0	1260.0	721.4 ^a,b^
2-ethyl-5-methylpyrazine	120.0	960.0	349.6 ^a^	170.0	1000.0	351.9 ^a^	330.0	770.0	448.0 ^a,b^	290.0	1210.0	705.6 ^b^	310.0	880.0	501.4 ^a,b^
2,3,5-trimethylpyrazine	100.0	2930.0	563.5 ^a,b^	250.0	1090.0	478.7 ^a^	440.0	1140.0	612.0 ^a,b^	380.0	1860.0	1001.1 ^b^	450.0	580.0	517.1 ^a,b^
2-ethyl-3-methylpyrazine	20.0	180.0	66.1 ^a^	30.0	160.0	65.0 ^a^	50.0	150.0	74.0 ^a,b^	40.0	260.0	118.9 ^b^	50.0	100.0	72.9 ^a,b^
2,6-diethylpyrazine	70.0	950.0	227.0 ^a,b^	90.0	530.0	198.1 ^a^	180.0	470.0	278.0 ^a,b^	150.0	780.0	410.0 ^b^	220.0	430.0	328.6 ^a,b^
3-ethyl-2,5-dimethylpyrazine	190.0	2100.0	682.2 ^a^	200.0	1720.0	616.3 ^a^	530.0	1590.0	906.0 ^a,b^	270.0	2770.0	1335.6 ^b^	520.0	1710.0	908.6 ^a,b^
PHENOLIC COMPOUNDS															
2-methoxyphenol (guaiacol)	270.0	1720.0	797.0	210.0	1290.0	656.3	470.0	1320.0	741.0	480.0	3010.0	1114.4	650.0	1560.0	978.6
2-methoxy-4-methylphenol	230.0	1320.0	597.0 ^a^	100.0	1010.0	456.9 ^a^	280.0	760.0	540.0 ^a^	450.0	1860.0	1055.6 ^b^	300.0	680.0	494.3 ^a^
4-ethylguaiacol	190.0	1610.0	697.0	180.0	1160.0	598.8	360.0	1350.0	677.0	400.0	2240.0	792.2	560.0	1440.0	930.0
2-methylphenol (*o*-cresol)	70.0	820.0	225.7 ^a,b^	40.0	360.0	168.8 ^a^	120.0	330.0	189.0 ^a,b^	160.0	750.0	338.9 ^b^	120.0	290.0	198.6 ^a,b^
3-methilphenol (*m*-cresol)	60.0	420.0	160.0 ^a,b^	30.0	170.0	103.1 ^a^	60.0	180.0	115.0 ^a^	110.0	620.0	248.9 ^b^	60.0	220.0	125.7 ^a^
p-vinylguaiacol (2-methoxy-4-vinylvinylphenol)	490.0	2720.0	1021.7	320.0	1610.0	772.5	510.0	1360.0	1008.0	680.0	2200.0	1221.1	400.0	2050.0	1120.0
SULFUR COMPOUNDS															
dimethylsulfide	nq	nq	nq	nq	nq	nq	nq	nq	nq	nq	nq	nq	nq	nq	nq
2-(methylsulfanylmethyl)furan (furfurylmethylsufide)	0.0	1270.0	424.3 ^a,b^	0.0	740.0	306.25 ^a^	0.0	970.0	524.0 ^a,b^	0.0	1140.0	730.0 ^b^	150.0	1000.0	477.1 ^a,b^
OTHERS															
unknown 1	220.0	3490.0	1200 ^a^	360.0	1690.0	901.9 ^a^	580.0	2840.0	1244.0 ^a^	1230.0	7150.0	2568.9 ^b^	320.0	1990.0	994.3 ^a^
benzylalcohol	120.0	520.0	237.8 ^a,b^	50.0	320.0	171.9 ^a^	150.0	310.0	213.0 ^a,b^	140.0	810.0	337.8 ^b^	170.0	280.0	231.4 ^a,b^
2-furanmethanol	740.0	6340.0	2628.3 ^a^	1170.0	6030.0	2415.0 ^a^	1560.0	4870.0	2808.0 ^a^	2320.0	8690.0	4891.1 ^b^	1830.0	4300.0	2831.4 ^a^
2-furanmethanolacetate (furfuryl alcohol acetate)	1160.0	8330.0	3427.4 ^a^	1170.0	3870.0	2829.4 ^a^	1790.0	7080.0	3843.0 ^a,b^	2870.0	11730.0	5692.2 ^b^	2260.0	6070.0	3641.4 ^a,b^
unknown 2	130.0	870.0	387.0 ^a^	80.0	850.0	320.6 ^a^	290.0	580.0	385.0 ^a^	310.0	1680.0	937.8 ^b^	280.0	730.0	421.4 ^a^
uknown 3	90.0	420.0	189.1 ^a,b^	30.0	260.0	142.5 ^a^	80.0	290.0	170.0 ^a,b^	60.0	470.0	271.1 ^b^	110.0	280.0	177.1 ^a^
furfurylammine derivative	230.0	1180.0	510.4 ^a^	100.0	700.0	359.4 ^a^	260.0	660.0	407.0 ^a^	360.0	1360.0	791.1 ^b^	300.0	1060.0	515.7 ^a,b^
unknown 4	70.0	360.0	153.0 ^a,b^	30.0	170.0	103.1 ^a^	60.0	180.0	114.0 ^a^	70.0	490.0	202.2 ^b^	70.0	240.0	131.4 ^a,b^
indole	40.0	230.0	95.2 ^a,b^	20.0	120.0	63.8 ^a^	40.0	120.0	72.0 ^a^	70.0	250.0	133.3 ^b^	30.0	110.0	71.4 ^a^
3-methylindole	50.0	350.0	135.7	30.0	240.0	94.4	40.0	190.0	114.0	70.0	380.0	153.3	40.0	240.0	145.7

**Table 3 molecules-25-01166-t003:** Inter-brand variation (%CV) in the content of volatile compounds detected in EC samples by chemical class.

	% CV				
CHEMICAL CLASS	BRAND A	BRAND B	BRAND C	BRAND D	BRAND E
ALDEHYDES	52.9	44.4	34.0	45.2	34.5
KETONES	54.6	43.2	35.6	50.7	35.1
ESTERS	50.0	40.1	37.4	48.0	35.9
TERPENS	42.4	34.8	34.8	51.8	19.0
PYRROLES	49.0	37.6	29.4	46.5	32.8
PYRIDINES	67.4	38.1	43.2	63.1	36.6
FURANS	51.1	38.2	44.0	53.9	28.9
PYRAZINES	70.2	49.3	35.1	48.9	30.0
PHENOLIC COMPOUNDS	52.3	44.5	27.0	56.2	32.0
SULFUR COMPOUNDS	74.4	37.3	52.0	69.7	53.8
OTHERS	54.7	32.6	35.6	44.5	31.1

**Table 4 molecules-25-01166-t004:** Key odorants detected in EC brewed by capsules.

**Compound**	**RI ^1^**	***m*/*z***	**Description of Odour ^2^**	**Typology of Contribution ^2^**
ALDEHYDES				
2-methylpropanale	827	41–72	Grassy, fermented	Negative
2-methylbutanale	919	57–58	Malt, fermented	Negative
3-methylbutanale	921	57–58	Almond, fruity	Positive
KETONES				
2,3-pentanedione	1063	57–100	Buttery, caramel like	Positive
PYRAZINE				
2-ethylpyrazine	1334	107–108	Earthy, musty	Negative
2-ethyl-6-methylpyrazine	1385	121–122	Earthy, musty	Negative
2-ethyl-3,5-dimethylpyrazine	1459	135–136	Woody, papery	Negative
PHENOLIC COMPUNDS				
2-methoxyphenol (guaiacol)	1867	109–124	Phenolic, spicy	Positive

^1^ Retention Index (RI) on SUPELCO^®^-Wax column, experimentally determined using homologous series of C8-C30 alkanes; ^2^ According to [1,15,17].

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
