# Peer review of "Chemical Characterization of Capsule-Brewed Espresso Coffee Aroma from the Most Widespread Italian Brands by HS-SPME/GC-MS"

_molecules, 2020, doi:10.3390/molecules25051166_

Round 1

Reviewer 1 Report

In this revised version, the manuscript is clear about its objectives. The inclusion of results (improvements in Table 2) and reformulation of the section "results" became more attractive for Molecules' readers.

Reviewer 2 Report

The manuscript does not require changes.

This manuscript is a resubmission of an earlier submission. The following is a list of the peer review reports and author responses from that submission.

Round 1

Reviewer 1 Report

Lolli et al. made an attempt to chemical characterization coffee capsules from five Italian brands. They used HS-SPME-GC-MS analyses supported by statistical and chemometric calculations. They determined about 70 volatile compounds in coffee probes.

In my opinion, this work is interesting and useful for understanding coffee composition. The manuscript is written correctly.

I suggest only two changes:

Please, include the Table S1 in the manuscript core Abstract, line 32 – abbreviation “PLS-DA” is not explained

Reviewer 2 Report

Although there are few references available on the chemistry composition specifically of espresso coffee capsules, volatile components of coffee are already known. Many articles explore and correlate various parameters involved from cultivars, processing and brewing methods. This manuscript propose to "...characterize the chemical aroma profile of EC capsules and to establish a correlation between the aroma profile of different coffee brands in capsules using chemometric tools". But the results presented are mere analyzes of the volatile composition of different capsule coffees. Some of the compounds or class of volatile compounds found are explored with chemometrics, but the relationship between these markers and the quality of the coffee drink is not evident. This, the authors themselves conclude in the conclusion "... that future studies are required".

Perhaps if the chemical composition related to the aroma of coffee obtained with HS-SPME was correlated with its sensory properties, that is, GC-MS could be used to define characteristics for quality control purposes. Capsules with specific sensory properties could be formulated according to a specific volatile profile. Chemometrics (multivariate analysis) could also be related to the sensory attributes of the capsules.